



# Assessment and comparison of thermal stabilisation measures at an Alpine permafrost site, Switzerland

Elizaveta Sharaborova[1,2], Michael Lehning[1,2], Nander Wever[2], Marcia Phillips[2], and Hendrik Huwald[1,2]

[1]Environmental Engineering Institute, Laboratory of Cryospheric Sciences, Ecole Polytechnique Fédérale de Lausanne (EPFL) Valais/Wallis, Sion, Switzerland
[2]WSL Institute for Snow and Avalanche Research SLF, Davos, Switzerland

**Correspondence:** Elizaveta Sharaborova (elizaveta.sharaborova@epfl.ch)

**Abstract.** Global warming provokes permafrost thawing, which leads to landscape changes and infrastructure damage, problems that have intensified worldwide in all permafrost regions. This study numerically investigates the impact of different thermal stabilization methods to prevent or delay permafrost thawing. To test different technical methods, an alpine mountain permafrost site with nearby infrastructure prone to damage is investigated. Model simulations represent the one-dimensional
effect of heat fluxes across the complex system of snow-ice-permafrost layers, and the impact of passive and active cooling, including engineered energy flux dynamics at the surface. Results show the efficiency of different passive, active, and combined thermal stabilisation methods, in influencing heat transfer, temperature distribution, and the seasonal active layer thickness. Investigating each component of thermal stabilization helps quantify the efficiency of each method and determine their optimal combination. Passive methods despite provide efficient cooling in winter, due to heat transfer to the atmosphere, are less
efficient as the active layer thickness remains over 1 m. Conductive heat flux regulation alone takes several years to form a stable frozen layer. Active, when powered with solar energy, cooling decreases the active layer thickness to a few decimetres. The combination of active and passive cooling, together with conductive heat flux regulation, performs best and allows excess energy to be fed into the local grid. Findings of this study show ground temperature and permafrost evolution at a representative alpine site under natural and thermally stabilized conditions, contributing to understanding potential and limitations of
stabilization systems and formulate recommendations for optimal application.

## 1 Introduction

The effect of increasing air temperature on permafrost is the result of complex, interacting processes occurring in different layers, including canopy, snow cover, and active layers of soil. On the continental scale, between 2007 and 2016, the Arctic continuous permafrost experienced warming of approximately 0.39 °C while the discontinuous permafrost warmed 0.20 °C
(Biskaborn et al., 2019). In particular, this warming reached 0.8 °C per decade at the Svalbard archipelago, and 0.5 °C per decade in Russia (Smith et al., 2022). In the same period, globally mountain permafrost temperatures increased by 0.19 °C (Biskaborn et al., 2019; Etzelmüller et al., 2020), responding to the raise of the atmospheric 0 °C isotherm (Kenner et al., 2024). Particularly the overall trend at some Alpine mountain permafrost sites warming at 10 m depth was 0.4 °C per decade between 1987 and 2009 (Haeberli and Gruber, 2009). These changes have led to an increase of the active layer thickness (ALT)



and shifted latitudinal permafrost limits northward (Biskaborn et al., 2019; Smith et al., 2022) and elevation limits to higher altitudes (Kenner et al., 2024). In Sweden, Greenland, and Svalbard the ALT has been increasing since the 1990s (Strand et al., 2021).

Also in mountain permafrost, ALT has been increasing since the 1990s, and in some places in the Swiss Alps it has even doubled in the 21st century (Smith et al., 2022; Swiss Permafrost Monitoring Network (PERMOS), 2024). At Schilthorn mountain in the Swiss Alps, annual average in-depth ground temperatures increased from -0.5 to +0.03 °C which resulted in doubling of the active layer depth (Swiss Permafrost Monitoring Network (PERMOS), 2024), (Hauck, 2002). In the Swiss Stockhorn borehole, where the rate of temperature rise increased by a factor 4, the decadal warming rate reached +0.64 °C for 2012–2022 (Morard et al., 2024). At Murtèl–Corvatsch, the permafrost warming over 18 years (1987-2006) was about 0.5 °C at 11.6 m depth and 0.3 °C at 21.6 m depth (Harris et al., 2009a). Over 35 years (1987-2022) of observation at Murtèl the temperatures at depth were increasing continuously, e.g., at 15 m to 20 m depth, from -1.8 °C to -1.0 °C (Haeberli et al., 2023).

Thawing permafrost and increasing ALT present a risk for built infrastructure. It has been estimated that almost 70% of such infrastructure might be affected by 2050, with transport infrastructure (roads, railways, cable cars, etc.) being the most vulnerable and most abundant (Hjort et al., 2018), (Hjort et al., 2022). A prominent Arctic example of infrastructure impacted by permafrost changes is the runway of the international airport at Longyearbyen, Svalbard, where the thawing of ice-rich soils caused uneven settlement during the summer season, and the autumn re-freezing provoked heave formation, eventually leading to runway destruction (Instanes and Mjureke, 2005). Besides transport infrastructure, over 1'000 settlements, and over 9'000 km of pipelines are located in areas subject to permafrost risk (Hjort et al., 2022). Failure of infrastructure might lead to environmental disasters as happened in May 2020 in Siberia, where 21'000 tonnes of oil were spilled in the Ambarnaya River causing devastating pollution with a total cost estimated at more than $2 billion (BBC News, 2020).

In mountain regions, such as the Swiss Alps, permafrost thawing also leads to noticeable impacts. In contrast to Arctic areas, mountain permafrost contains less excess ice. This small amount of ice acts like a natural "glue", stabilising slopes and cliffs. Its thawing weakens the static stability and therefore increases the risk of rockfalls, landslides, and debris flows (Gruber and Haeberli, 2009), (Noetzli et al., 2003). Infrastructure such as cable cars, houses, antennas, avalanche protection systems etc. are put in danger, due to the mentioned risk and other destruction directly caused by permafrost thawing (Duvillard et al., 2021). An example is the 2017 large landslide in Bondo, Switzerland (Mergili et al., 2020), a compound event of cascading processes involving progression of a rockfall, to rock (or rock-ice) avalanche and to a debris flow. Such processes chains are often directly related to changes in the cryosphere, e.g., glacial de-buttressing, retreating hanging glaciers, or vanishing permafrost (Harris et al., 2009b; Haeberli et al., 2017).

All these challenges related to new or increased natural hazards and to building on mountain permafrost, require technical adaptation, including the use of flexible or semi-mobile structures, advanced materials and cooling techniques (Haeberli et al., 2010). Finding solutions for the stabilisation of permafrost substrates in the currently affected regions, above, below, or around critical infrastructures, and wherever new constructions on permafrost are planned is becoming inevitable. Stabilisation of permafrost means primarily keeping it at sub-freezing temperatures and minimizing the depth and dynamics of the ALT.





There are several different thermal stabilisation or ground cooling methods currently applied worldwide. They can be divided

into passive methods, i.e., these not requiring external energy during operation, and active methods needing extra power supply.
Common passive thermal stabilisation systems are based on the principle of increasing thermal resistance in the upper layer
of the ground. Among these methods are solar reflectors and shields (Qin et al., 2020), thermal insulation (Luo et al., 2018),
and additional layers on top of the ground for protecting from solar radiation and net heat accumulation (Yinfei et al., 2016).
The aim of these methods is to reduce the amount of net solar radiation, heat convection, and heat conduction above, within,

and around the affected surface. However, these methods lack long-term effectiveness. To improve them, it has been recently
proposed to regulate the heat transfer by insulating the ground in summer to prevent heat penetration, and lifting the insulation
in winter to allow effective ground cooling (Sharaborova and Loktionov, 2022). Other techniques based on lowering the ground
temperatures (Cheng, 2005) include so-called thermosyphones. The technology uses a sealed tube filled with a fluid (e.g.,
carbon dioxide, ammonia, or propane) to transfer heat from the soil to the atmosphere. The tube has two parts: an evaporator in

the ground and a radiator above ground. When the ambient air temperature drops below ground temperature, vapour condenses
in the radiator, reducing pressure and causing liquid in the evaporator to boil and transfer heat to the radiator. The objective
of thermosyphones is to lower the ground temperature through freezing during winter time and to ensure low temperatures
during summer. Thermosyphones were initially proposed for freezing ground during wintertime beneath foundation pillars
located approximately 6 meters below the seasonally thawed layer. This technology was also tested for protecting underground

infrastructure (tunnels) in permafrost zones (Zhang et al., 2017). A variation of thermosyphones was effectively implemented
into highway embankments using heat pipes (Tian et al., 2021). However, in some particular cases and for certain slope
orientations there is still a risk of surface deformation. In addition, due to the climate change they are loosing their efficiency,
because the winter becomes shorter and the frozen volume is not large enough to keep the ground stable.

In recent years, active cooling methods have undergone significant development, but their high cost due to the demand for

substantial power has limited their widespread use. Some progress has been made in the direction of using deep cooling in an
active mode using solar energy (Hu et al., 2020). Their experiments demonstrated that active cooling is suitable for permafrost
protection and that utilizing solar energy offers an economical solution for achieving field refrigeration without relying on
grid power in permafrost regions. This method could be improved upon by shielding from solar radiation and using efficient
energy redistribution and utilization. It has been shown in (Asanov and Loktionov, 2018; Loktionov et al., 2019), that utilizing

solar power panels installed on embankments helps to achieve the combined effect of shielding from direct solar radiation and
powering the cooling system. This method is based on a combination of passive and active thermal stabilisation techniques
(Sharaborova et al., 2022a), where the solar panels act as sun screens, while simultaneously power the heat pump being the
active component in the cooling process. The system uses pipes below the ground surface in which cooling liquid is circulating
creating a cold (sub-zero) thermal layer, often referred to as "barrier layer" preventing heat from the surface being conducted

deeper into the ground. The barrier layer, as introduced by (Sharaborova et al., 2022a; Loktionov et al., 2022), serves to shield
cold ground against ambient heat penetration, unlike thermosyphons which cool massive volumes in depth. To achieve this,
cooling pipes are placed parallel to the ground surface within the natural thawing layer near the surface. The effect of this
method has been simulated for the lowland Siberian permafrost (Loktionov et al., 2022) using a 3D finite element method



(FEM) package, specifically developed for permafrost soil calculations (Alekseev et al., 2018), and it has been experimentally

tested (Sharaborova et al., 2022b).

Permafrost in mountain regions poses extra challenges due to the settlement, deformation, and creep of the ground that can occur in complex, ice-bearing terrain, requiring careful analysis, assessment, exploration of solutions, and timely remedial measures to ensure maximum infrastructure lifetime (Bommer et al., 2010). There are special construction methods that can be applied for mountain buildings, like cable cars and restaurants subject to deformation (Bommer et al., 2010). For example,

a technique with flexible foundation systems can be used to carry out geometrical corrections when infrastructure subsides or creeps (Phillips et al., 2007; Harris et al., 2009a). Thermal stabilisation methods, including insulation to mitigate heat transfer with air spaces or pressure-resistant materials like foam glass or extruded polystyrene, and passive cooling systems like thermosyphons are utilised in mountain permafrost construction, though the latter are not widespread in the Alps (Phillips et al., 2007; Bommer et al., 2010).

To test and estimate the effect of thermal stabilisation methods on-site, experiments and monitoring are required. However, experimental designs are time consuming and expensive. Modelling using numerical simulations is a good alternative to examine the impact of cooling methods and to provide a better understanding of thermal changes occurring in permafrost. Different models are used to simulate permafrost. Established physics-based 1D models such as the CoupModel (Schaefer et al., 2014; Jansson and Karlberg, 2004; Marmy et al., 2013, 2016) and SNOWPACK (Lehning et al., 1999) are used to simulate mass and

energy exchange processes in the soil-snow-atmosphere system. Such models have also been extended to cover wider areas as 1D distributed columns, as is the case for Alpine3D, which uses the SNOWPACK model (Haberkorn et al.) or a recent update of GERM (Huss et al., 2008; Farinotti et al., 2012) with a permafrost module (Pruessner et al., 2021). These models can also be used to simulate protective insulation methods. An example is the SNOWPACK simulation of a partial glacier protection using a geotextile cover (Olefs and Lehning, 2010) which allows to examine in detail the energy exchange between different layers.

Another example is the use of the above-mentioned 3D FEM package to gauge the effectiveness of a thermal stabilisation method in permafrost (Loktionov et al., 2022).

The present study uses the SNOWPACK model to investigate the processes occurring in permafrost during the application of passive and active thermal stabilisation methods at a selected mountain permafrost site. The principal objectives are (1) to numerically examine the heat flux distribution in complex snow–ice–permafrost substrates and on the contact border with

thermal stabilisation systems, (2) to quantify the impact of thermal stabilisation on the active layer thickness (ALT). We investigate how various cooling methods affect the long-term evolution of permafrost and its preservation in a cold state and how they influence the thickness of the active layer, in combination with thermal stabilisation approaches. The results are expected to help to assess the pros and cons of different thermal stabilisation methods for permafrost protection and preservation. The numerical studies allow the creation of an integrated view of the technology behaviour and help to adapt it

for in-situ application.

This study focuses on numerically evaluating the performance of thermal stabilization methods using borehole data at Schilthorn, Switzerland, a high-altitude site representative of alpine permafrost regions. By analysing their impact on key variables such as ground temperature, heat fluxes, and ALT, we aim to provide insights into the efficacy of these measures and





their potential of improving the resilience of permafrost against atmospheric warming. The findings contribute to optimizing
engineering practices in permafrost regions and addressing knowledge gaps in permafrost thermal management.

## 2   Modelling with SNOWPACK

Here, we build upon previous research employing the SNOWPACK model for permafrost studies (Pruessner et al., 2021, 2018;
Luethi et al., 2017). The model was initially developed for avalanche warning (Lehning et al., 2000) and it computes the
1D heat transfer, water transport, vapour diffusion, and mechanical deformation, including new snow, wind drift, and snow
ablation. It includes a description of soil layers (Luetschg et al., 2008) where the physical properties (thermal conductivity,
heat capacity, density) remain constant over time (Pruessner et al., 2021). SNOWPACK includes both a simple bucket water
transport model (Lütschg, 2005) and an advanced Richards water transport model (Wever et al., 2014, 2015) to simulate water
transfer from the snow cover to the underlying substrate. In the current study, we use a simple bucket scheme, which allows
us to set the fixed substrate volumetric content for the layers and not to specify the grain size of the soil layers to determine
the water retention properties of the soil. The bucket model in SNOWPACK is computationally very efficient and was tested
for various mountain permafrost sites. SNOWPACK also allows the incorporation of artificial materials on the surface as an
additional layer. This feature has been used to successfully model geotextile-covered snow (Olefs and Lehning, 2010) showing
a good agreement between the modelled and experimental observed temperature profile of artificially conserved snow. This
model feature is also used in the present work to represent the inclusion of thermal insulation material.

Heat sinks and -sources can be represented in the model by an advective heat flux formulation (Luethi et al., 2017). This
capability can be used to heat or cool any layer, including the soil, and model, for example, talik formation in the permafrost
(Luethi et al., 2017). Based on these previous applications of SNOWPACK, the model is considered well suitable for studying
the effects of thermal stabilisation systems in mountain permafrost. In this study, we used SNOWPACK Version 3.7 to model
the effects of cooling pipes in the ground. We introduced an optional feature to the model to allow for a representation of the
artificial cooling as if it was installed on the site, and to control heat advection from the pipes. The modified version of model
includes a switch to regulate temperature and remove the heat from the ground by activation the advective heat mode. The user
sets a temperature limit. When the ground temperature reaches set threshold, the cooling system turns off, in the model it means
advective heat is deactivated. Otherwise, if the temperature of the ground is too high - above the established threshold - cooling
is activated. However, due to the 1-D formulation of the model, exact simulation of the temperature along the cooling pipes
is not possible. The model prescribes with a constant temperature in the pipe, not considering the fluid temperature difference
at the entrance and exit of the pipe due to heat conduction to the surrounding ground as would be the case in a 3-dimensional
situation. Another approximation is that the calculation of the power required for cooling is simplified, and only effective when
solar radiation is available. A more advanced way would be to monitor the cooling, and adapt the power supply according to
the conditions of the ground and the solar power production. The current state of the model allows us to assess the thermal
stabilisation potential of a system and show its effects when applied at a permafrost site. However, for a more detailed study



of the stabilisation application further model refinement is still required. This includes minimizing the current approximations and simplifications, aiming for a representation that better captures practical behaviour.

The rational of using SNOWPACK for modelling thermal stabilisation in permafrost compared to previous work (Loktionov et al., 2022) is the better representation of snow, ground parameters, and the permafrost temperatures during the entire sim-
ulation. In Loktionov et al. (2022), a 3D model was used with third-type boundary conditions at the soil surface, where heat fluxes were calculated using air temperature and heat transfer coefficients based on wind speed. However, in SNOWPACK, boundary conditions can be set using Neumann conditions, directly calculating the heat fluxes at the surface. It also avoids the connection with surface temperature, which will be changing with the application of the cooling methods. These boundary conditions provides a more accurate representation of the heat exchange between the ground and the atmosphere, with a better
representation of the heat flux at the soil-atmosphere interface. Even if the model from Loktionov et al. (2022) uses the third-type of boundary condition, which can be benefit for permafrost modelling, the advantage of SNOWPACK is that it presents a full set of surface fluxes, including a wide choice of stability corrections for turbulent fluxes. In addition, SNOWPACK enables the detailed representation of every model layer including temperature, heat flux, and the possibility of monitoring the output of the permafrost parameters. Another advantage is the possibility of modelling multi-year periods in very short time,
the inclusion of radiation effects on the ground, and the possibility of controlling the albedo effect.

## 3   Study site and data

In this study, we use borehole temperature data from the PERMOS mountain permafrost site Schilthorn, canton Bern, Switzerland (46.558292 N, 7.834626 E, 2923 m a.s.l.)(Swiss Permafrost Monitoring Network (PERMOS), 2024). This site is representative for many locations of alpine permafrost and because the nearby infrastructure will be affected by permafrost thaw.
Over the past decade, the ALT at this site has doubled (Hauck and Hilbich, 2024). By studying Schilthorn during a period when the ALT was thinner and before these significant changes occurred, we gain valuable insights into the dynamics of mountain permafrost. The ALT thickening causes direct risks to infrastructure, such as cable car foundation, hiking trails, and mountain huts. Permafrost thaw can lead to slope instabilities, rockfalls, or subsidence. Indirectly, this results in economic losses due to repair costs and danger to recreation activities. These risks are especially significant for infrastructure located on steep or
unstable slopes. These insights are also applicable to similar Alpine sites with cable car infrastructures or other facilities, such as the Zermatt/Matterhorn area in Switzerland (Weber et al., 2017), or the Chamonix/Mont Blanc region in France (Duvillard et al., 2021).

Figure 1 shows the location of the site with respect to the spatial permafrost distribution and solar radiation level. Meteorological and borehole stations from Table 1 are shown in Figure 1 on the right. The observational permafrost temperature and
atmospheric data sets of this site are largely sufficient for modelling purposes.

The borehole temperatures were obtained from the Schilthorn SCH_5198 borehole operated by PERMOS (Swiss Permafrost Monitoring Network (PERMOS), 2024). The borehole was established in 1998, it is 13 m deep and is still active. For the simulations, we used atmospheric data from the PERMOS meteorological station next to the borehole (Hoelzle et al., 2022;





**Figure 1.** Map of Switzerland indicating the Schilthorn study site (red triangle). The map shows permafrost distribution adapted from (Kenner, 2018) (spectrum from dark blue to light blue), and yearly averaged solar radiation (Solargis, 2020) (spectrum from blue to red). The site is shown on the bottom left (Hilbich, 2010) and the stations' (green circles) position from Table 1 on the right.

Swiss Permafrost Monitoring Network (PERMOS) and Hoelzle, 2021). Due to the presence of gaps the data was completed
with data of the nearest Intercantonal Measurement and Information System (IMIS) stations and MeteoSwiss meteorological
stations (Table 1). To reduce the bias related to the used input data, correlations between the Schilthorn meteorological station



**Table 1.** Meteorological and borehole data used for modelling.

| Station | Coordinates | Data used |
|---------|-------------|-----------|
| Schilthorn borehole (Swiss Permafrost Monitoring Network (PERMOS), 2024) | 46.558 N, 7.835 E 2900 m a.s.l. | Borehole temperatures at depths (m): 0.2, 0.4, 0.8, 1.2, 1.6, 2, 2.5, 3, 3.5, 4, 5, 7, 9, 10, 11, 13 |
| Schilthorn meteo station (Hoelzle et al., 2022; Swiss Permafrost Monitoring Network (PERMOS) and Hoelzle, 2021) | 46.558 N, 7.835 E 2900 m a.s.l. | Air and surface temperature, relative humidity, wind speed, radiation (all components), snow depth |
| Schilthorn IMIS | 46.557 N, 7.835 E 2996 m a.s.l. | Air temperature, wind speed, relative humidity |
| Türliboden IMIS | 46.577 N, 7.835 E 2332 m a.s.l. | Reflected short-wave radiation, surface temperature, snow height, precipitation |
| Gipfel MeteoSwiss | 46.557 N, 7.835 E 2970 m a.s.l. | Wind speed |

IMIS: Intercantonal Measurement and Information System

and the other stations were calculated and existing gaps were filled based on the obtained regression coefficients. The remaining gaps were filled with the MeteoIO interpolation and generation functions (Bavay and Egger, 2014).

We set-up the model to correspond to observational data using the following set of parameters: volumetric ice, water, substrate, and voids content of the ground; substrate density, thermal conductivity, and heat capacity. The site is free of vegetation and the substrate consists of deeply weathered dark limestone at the surface of mainly sandy and gravelly debris of several meters depth, which lies over strongly jointed bedrock (Zappone and Kissling, 2021). The required initial conditions for the model were chosen to represent rocky ground, without specifying particular grain sizes for different layers. Based on the available borehole temperature measurements data and previous modelling experience (Hoelzle et al., 2022; Marmy et al., 2013; Ekici et al., 2015; Wagner et al., 2019), it is assumed that the substrate parameters are those shown in Table 2. The top layer consists of well-drained loose rock (talus) and thus has a lower water content and moderate porosity, which explains the lower heat capacity required to reproduce temperature distribution. Below this, the heat capacity increases representing the presence of an active layer, which is indicated by long spring and autumn zero-curtains (Swiss Permafrost Monitoring Network (PERMOS), 2024). The zero-curtain effect happens due to the slowed latent heat release during the phase transition (thawing or freezing) in the active layer. At this time the ground temperature remains near 0°C over an extended period, meanwhile, the air temperature may vary. At depth, the standard values for limestone are used. The values for density and thermal conductivity are according to the given substrate stratification. The geological map (Zappone and Kissling, 2021) indicates a limestone bedrock on this site in depth and the with sandy and gravelly debris in upper layers, justifying the use of the corresponding values recommended for this type of ground (Table 2).





**Table 2.** Parameters of geological substrate for the Schilthorn site simulations.

| Depth, m | Volumetric content (ice, water, void, substrate), % | Density, kgm$^{-3}$ | Thermal conductivity, W m$^{-1}$ K$^{-1}$ | Heat capacity, J kg$^{-1}$ K$^{-1}$ |
|---|---|---|---|---|
| 0-2.5 | 0; 20; 21; 59 | 1600 | 2.2 | 700 |
| 2.5-5 | 5; 20; 6; 69 | 2000 | 2.7 | 1000 |
| 5-13 | 10; 0; 0; 90 | 2700 | 3.0 | 900 |

## 4 Model simulations

To understand the processes happening in permafrost soils, different configurations of numerical simulations were employed, first simulating natural conditions without engineered modifications and then applying different thermal stabilisation methods. The simulations have been run from June 2000 until January 2017. The meteorological forcing at 2 m height and the model output both have an hourly time step.

### 4.1 Natural conditions

Numerical simulations of natural conditions were carried out as a reference case and to evaluate the performance of the model compared to the measured temperatures from the borehole. These simulations, combined with the study of the modelled and measured substrate parameters at this site (Pellet et al., 2016; Scherler et al., 2010; Marmy et al., 2016), allowed for selecting the right setting for optimizing the agreement of the SNOWPACK modelled temperature distribution and the measured borehole temperature data. Albedo was set to 0.15 for soil and we used the Schmucki et al. (2014) albedo parametrization for snow. The roughness length was set to 0.05 m owing to the rocky surface. We used the Holtslag and Bruin (1988) correction model for atmospheric stability, which shows a good performance over snow surfaces (Schlögl et al., 2017). To facilitate modifications during the thermal stabilization experiments we force the model with incoming short-wave radiation mode, while the other components are computed by the model based on available meteorological forcing data (air temperature, wind speed, relative humidity, incoming short-wave radiation, snow height, liquid precipitation equivalent and bottom ground temperature) for a full surface energy balance assessment. At the surface, a Neumann boundary condition has thus been applied. The surface temperature data (Table 1) is not directly used as input for the model simulations but serves for comparison of the model output to the measurements for consistency. At the base of the simulation domain, a Dirichlet boundary condition is used, prescribing borehole measurements ranging from -0.9 °C to -0.1 °C.

We quantitatively compared the model output with borehole temperature measurements using standard statistical analysis such as bias, root mean square error (RMSE), and correlation (Pearson coefficient) at different depths for monthly mean averages and for the whole time series. Additionally, we compared mean values and standard deviation for different model setups with observations. The best performance setup was chosen as a reference case.





## 4.2 Thermal stabilisation experiments

### 4.2.1 Shading of the surface

Surface shading protects the soil from direct solar radiation, snow, and liquid precipitation. It also affects the wind speed near the soil. The effect of the snow accumulation is that it works as a natural insulation, therefore, in winter it doesn't let the heat be extracted from the ground, and limits natural cooling (Wang et al., 2019). As for the precipitation, it negatively affects by increasing the ALT which accelerates permafrost degradation (Zhu et al., 2017; Wang et al., 2021). Wind speed decreasing/increasing can affect as positively as negatively on permafrost, impacting the heat transfer, snow redistribution and moisture content, which in turn can either protect or accelerate permafrost degradation (Zhao and Yang, 2022). We simulated shading using solar panels positioned above ground level, exceeding the height of the snow cover, as proposed by (Loktionov et al., 2022). To quantify the impact of shading from various parameters individually and combined, we conducted experiments in two stages. In the first stage, we investigated the effect of each of the following parameters independently from each other: (1) wind speed decreased by 70%; (2) ground completely protected from snow and liquid precipitation; (3) no direct component of the incident short-wave radiation leaving only the diffuse component, which is derived from the reference case SNOWPACK simulation under natural conditions . The choice of these parameters was partly based on previous experiences from modelling the thermal stabilization effect (Loktionov et al., 2022). However, in this study the solar radiation was decreased by 95% from global horizontal irradiance (GHI). In the present study we can revert to the modelled components of direct and diffuse solar radiation. In the case of the ground being shaded by the solar panels, we are neglecting the direct radiation component. Using only the diffuse solar radiation component is a simplified and idealised case, which allows to reproduce the effect of the solar panels on ground temperature distribution, without applying an additional empirical coefficient. Removal of the direct solar radiation component reduced the radiative heat flux from the atmosphere accordingly(Liu and Jordan, 1960; Olson and Rupper, 2019; Li et al., 2020). The real effect of the solar panels may bring a bigger/smaller impact on the solar radiation intensity reaching the ground beneath solar panels. The numerical tests with these assumptions allow to model the impact from the solar panels placed above the ground, however, for more accurate analysis, in-situ experiments are required. Test results will show the impact of each change on thermal stabilisation, its efficiency, and the seasonality. During the second stage, all parameters were modified simultaneously. This experiment simulates the total effect of the heat transfer and vertical temperature distribution with the application of the shading method. The objective of this experiment is to allow more efficient surface cooling in winter by heat loss to the atmosphere and by reflecting solar radiation.

### 4.2.2 Thermal insulation

We also consider the situation of the ground being covered with thermal insulation material. We simulate the presence of a 50-100 mm thick polystyrene slab with a thermal conductivity of $0.033 \, \mathrm{W \, m^{-1} \, K^{-1}}$, a density of $28 \, \mathrm{kg \, m^{-3}}$ and a heat capacity of $1500 \, \mathrm{J \, kg^{-1} \, K^{-1}}$ to reduce heat transfer from the atmosphere to the ground (Loktionov et al., 2022). Test simulations showed that a thickness of 50-100 mm is needed to have a sufficient insulating effect. It additionally increases the albedo, reflecting more solar radiation from the ground. This was implemented in the model as an additional layer at the top of the soil to preserve





the cold at depth. The albedo of the isolation material is assumed to be 70% similar to a snow cover. Two particular cases are investigated: (1) insulation present all year round, and (2) insulation placed on the ground from the beginning of June until

the end of October (under snow-free conditions) and lifted from November until the end of May to favour the natural cooling process. In (1), the insulating material protects the ground from direct contact with snow and rain. However, both precipitation types still continue to accumulate on top of the protection layer which is considered in the simulation. At this step, we also studied the impact of thermal insulation thickness, by showing different cases of thermal insulation material thickness (50 mm, 100 mm). The objective of (2) is monitoring the heat flux regulation (Sharaborova and Loktionov, 2022) by favouring natural

cooling in winter and preserving the low soil temperatures in summer. The advantage of (2) is the combination of the shading effect of the solar panels (see Section 5.2.1) with ground insulation material. This term has been introduced in the method description (Sharaborova and Loktionov, 2022), and relates to the regulation of conductive heat flux and temperature. The heat flux regulation effect is achieved in this invention for the prevention of permafrost thawing without external power supply. The method stabilises frozen rocks by insulating the soil in warm seasons by protecting the ground from solar radiation, and

precipitation, and limiting convective heat transfer. Elevating the insulation above snow cover in winter helps to use natural air circulation for cooling.

### 4.2.3 Thermal stabilisation with active cooling

This experiment simulates the case of thermal stabilisation using a solar-powered heat pump (Sharaborova et al., 2022a). The system combines the previously described passive protection (Section 4.2.1), where the solar panels act as sunscreens, and at

290 the same time also power the heat pump for cooling the ground through cooling pipes. The latter are modelled by implementing an sink term which corresponds to the energy produced by solar panels and available for cooling the soil by means of the heat pump and keeping the temperatures around ground pipes at temperatures lower than $0\,^{\circ}\mathrm{C}$.

Based on previous studies modelling a thermal stabilisation system application (Loktionov et al., 2022), we model the cooling pipes by applying a negative advective heat ($q_{\mathrm{adv}}$) in the concerned layers below ground. The advective heat is implemented

in the following way: First, the energy obtained from the solar panel is calculated using the incoming short-wave radiation (ISWR), the solar panel surface area ($A = 1\,\mathrm{m}^2$), and the photovoltaic (PV) conversion efficiency ($\eta = 10\%$), as detailed in Equation (1). Then, this power is applied as cooling power, using the cooling energy efficiency rate (EER), Equation (1), which is related to the coefficient of performance (COP) for cooling machines and for heat pumps, according to Equation (2), and depends on the ambient (atmospheric) temperature, as evident in Equation (1). Equation (1) is adopted from the previous study

of this thermal stabilisation method (Loktionov et al., 2022), the coefficients there are obtained based on the characteristics of the equipment. This relation of parameters is different depending on the model and design of the heat pump. Finally to determine the advective heat that is applied in the model the cooling capacity is divided by the volume of the cooling application. This procedure is described using the following equations:



$$q_{\text{adv. heat}} = \frac{\text{ISWR} \cdot \eta \cdot A \cdot \text{EER}}{V} \tag{1}$$

$$\text{EER} = 4.8 - 0.12 \cdot T_a \tag{2}$$

$$\text{EER} = \text{COP} - 1 \tag{3}$$

The advective cooling was applied in the model for the months of April to October. The cooling pipes are integrated into the model as advective cooling at a layer from 20 cm to 22.5 cm (the diameter of the pipe is 25 mm), and are cooling this layer to a minimum temperature of $-7.5\,^\circ$C, which corresponds to the mean temperature of coolant expected in the pipes. The active cooling from the pipes creates a barrier layer of a cold (frozen) slab inside the ground protecting the permafrost in summer from heat conducted from the surface deeper into the ground. Experiment 1 applies active cooling with pipes only but without shading the soil with solar panels. Experiment 2 combined both elements, i.e., passive and active cooling. We consider that the surface of solar panels is equivalent to the cooling surface as 1:1, in both experiments. However, in real life, this relation will be more complex, depending on the time of the day, slope orientation, ground parameters etc. Later in this study, we examined also the cases with partial energy use from the solar panel (Section 5.3), when only part of the energy (from the total production by solar panels) is reserved for the cooling.

## 5 Results and discussion

Before discussing the specific results of the different simulations, we want to reiterate the main objective of this study being the evaluation of the performance of engineering methods for thermal stabilisation of alpine permafrost and its sensitivity on different parameters. Also note that the results are specific for the particular site conditions at Schilthorn, although they can be considered representative for a range of conditions typical for alpine permafrost regions, and that parameter calibration was carried out to best reproduce the measured temperature profile evolution for the natural situation.

### 5.1 Natural conditions

The simulated ground temperature evolution together with the snow depth at the Schilthorn site is shown in Figure 2 for the entire observation period from 2000 to 2017. The seasonal active layer outlined with the $0\,^\circ$C isotherm is evident and the ground temperature at depth corresponds to the measured permafrost temperature in the borehole. We can also note an increase of the ALT over years from about 5 m to almost 8 m. In natural conditions, realistic computation of the ground heat flux follows from an accurate representation of the snow height as shown in Figure A1, where modelled and measured snow align closely for most of the winters. For some periods, measured snow depth is below the simulated one indicating that the model would underestimate melt or erosion. Figure 3a indicates that the modelled temperatures are slightly higher than measured. However, the monthly averaged difference for the whole period of simulation never exceeds $1\,^\circ$C. Figure 3b indicates that the model and the observations are comparable with similar long-term mean temperatures and standard deviations in all model layers evidenced by the correlation coefficients larger than 0.8 in all layers, except for the middle layers, notably at the depth around



5 m, where the coefficient drops to almost 0.6. This depth corresponds to the ALT where the ALT sensitively depends on small

changes in temperature (on the order of a tenth of a degree) resulting in relatively large fluctuations in the position of the ALT (Marmy et al., 2016).

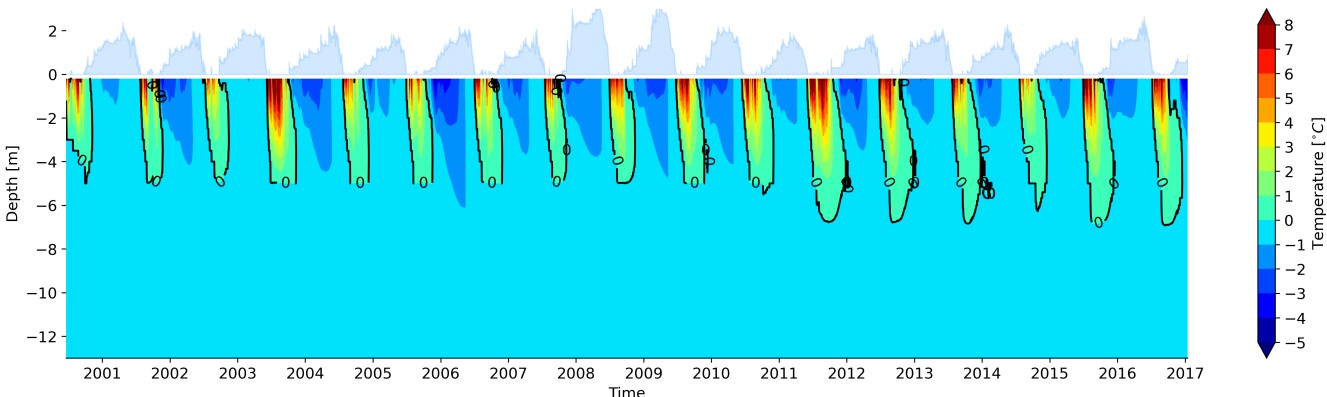

**Figure 2.** Time series of daily averaged modelled ground temperatures during natural undisturbed conditions at the Schilthorn site from 2000 to 2017. Black contours indicate the 0 °C isotherm, i.e, the ALT. Snow depth is indicated in light blue above the 0 m depth level. The 0 °C isotherm is outlined as black contour.

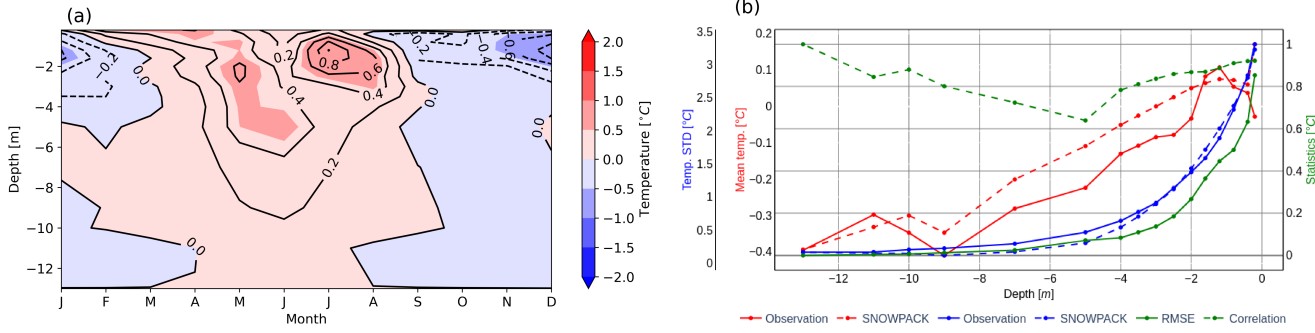

**Figure 3.** Comparison between modelled and measured ground temperatures at the Schilthorn site from 2000 to 2017. (a) Monthly averaged differences. Red (blue) colours indicate model temperature higher (lower) than observations. The solid (dashed) black contour lines represent positive (negative) isotherms. (b) Statistical analysis of the ground temperature differences averaged over the 17 years period. The 3 vertical axes show: mean temperature (red), standard deviation (blue), other statistics (green). The horizontal axis indicates depth.





**Figure 4.** Monthly averaged modelled ground temperatures at the Schilthorn site from 2000 to 2017 resulting from individual model configurations defined in the "shading" experiment (Section 4.2.1). Panels (a, c, e, g): simulated ground temperature, panels (b, d, f, h): difference between simulated and natural conditions. Red (blue) colours indicate model temperature higher (lower) than observations. Experiments shown: (a, b) - wind speed reduced by 70%; (c, d) - liquid precipitation set equal to zero; (e, f) - snow depth set equal to zero; (g, h) - direct component of incoming short-wave radiation neglected.





## 5.2 Thermal stabilisation experiments

### 5.2.1 Shading of the surface

We studied the impact of shading on the ground temperature in several experiments isolating the effect of individual elements as described in Section 4.2.1 by comparing the simulations with the averaged natural conditions (Figure A2). Modification of liquid precipitation by sheltering effects results in the smallest temperature changes in the ground (Figures 4d). The absence of liquid precipitation leads to better cooling in winter, and reduction of the ALT thickness. Decreasing the wind speed results in more efficient cooling for the full year (Figure 4a,b) as a result of changes in sensible heat fluxes over the surface. The biggest

effect occurs when the snow depth is reduced to zero (Figures 4e,f). This change results in a remarkable cooling of the ground in winter (Figure 4e) due to the removal of the natural insulation layer. This effect helps to cool permafrost during winter. However, temperatures in the summer season increase since during the usual melting season the absence of snow is no longer protecting the ground from heat and the favourable albedo effect is missing. If no snow is accumulated the benefit of the natural insulation is lost and the warming of the ground starts earlier than otherwise (Figure 4f). In that case, the lack of snow during

the melting season affects negatively the ground temperatures, and the heat taken from the ground in winter is not enough to compensate. Finally, shading the ground from the incoming direct short-wave radiation (Figures 4g,h) results in decreasing the temperatures of the ground, in winter and, especially, in the summer season. This occurs because the energy received through radiation is not penetrating the ground and, as a result, the depth of the thawing layer (0 °C) is less important.

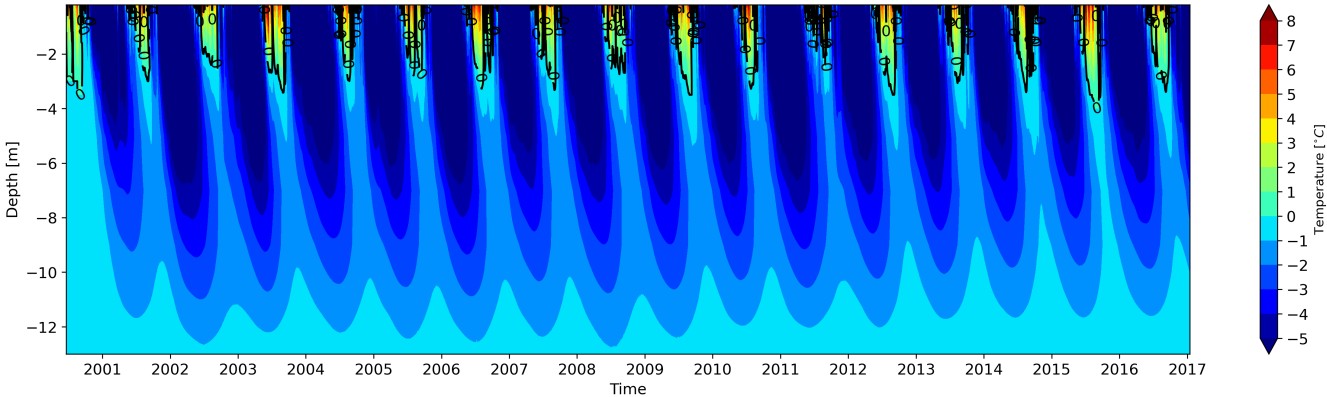

**Figure 5.** Time series of daily averaged modelled ground temperatures from the shading experiment at the Schilthorn site for the period of 2000 to 2017. Black contours indicate the 0 °C isotherm, i.e, the ALT.

In Figure 5 the total effect of shading at Schilthorn is shown, demonstrating that the shading leads to cooling of the ground

during winter and decreases the ALT (depicted as the 0 °C isotherm in both figures). However, there are positive temperature differences in the summer season, which provokes ALT remain at certain meters depth during the evaluation time, indicating that this method of stabilisation is insufficient and should be supplemented by another, as was mentioned by (Cheng, 2005) and (Loktionov et al., 2022). This method cannot stabilise the cold region below the surface. On the contrary, it amplifies the annual





cycle of thawing and refreezing. In the long-term trend (Figure 5), the net effect of this method allows rapid ground cooling in
the winter every year, however, it does not create and maintain a thermal barrier layer even after 16 years in application, and
consequently, the atmospheric heat continues to affect the ground temperatures.

### 5.2.2 Thermal insulation

In Figure 6, the effect from year-round artificial thermal insulation layer is shown, presenting the monthly averaged evolution
of the ground temperature profile of the Schilthorn borehole. We examine two cases with different thermal insulation layer
thicknesses of 50 mm and 100 mm. While the placement of an insulation layer of 50 mm does not completely prevent temper-
atures above 0 °C (Figure 6a) it decreases the ALT in summer (Figure 6b). In addition, Figures 6b and B1 demonstrate that the
placement of an extra artificial insulation layer decreases the heat loss to the atmosphere in winter. The heat flux distribution
in the permafrost remains preserved, due to the heat flux limitation at the surface (Figure B1). However, on the long-term the
combination of the atmospheric heat flux together with the geothermal heat flux will warm the permafrost. Furthermore, the
deployment of an insulation layer of 50 mm thickness at the surface is still insufficient to preserve the permafrost table, and
the ALT still comprises several meters (Figure 6a).

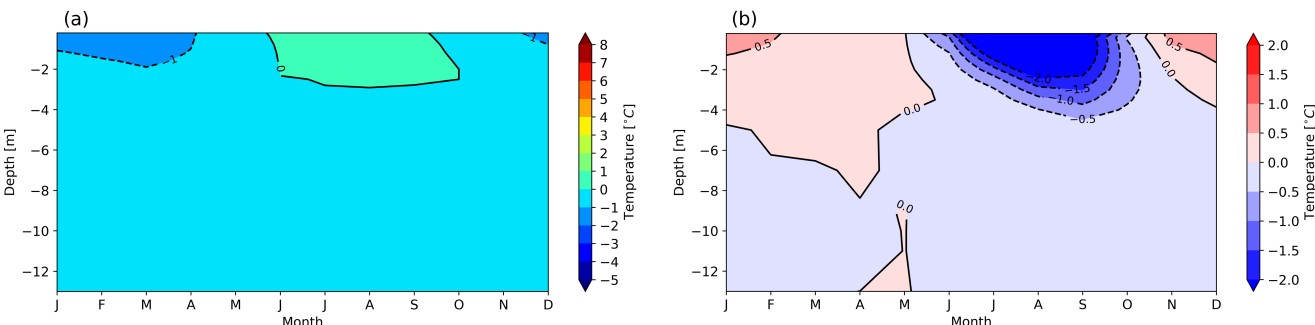

**Figure 6.** Modelled monthly averaged ground temperatures at the Schilthorn site for the period from 2000 to 2017 using a 50 mm artificial
thermal insulation layer on top of the ground all year round. (a) Temperature evolution in the ground. Black contours indicate the 0 °C
isotherm. (b) Temperature difference compared to the undisturbed natural conditions. Red (blue) colours indicate model temperature higher
(lower) than measurements. Solid (dashed) black contour lines represent positive (negative) isotherms.

Results in Figure B2 show that the thicker the thermal insulation layer the smaller its effect of thermal stabilisation. Instead
of favouring cooling of the ground during winter, the artificial thermal insulation prevents the desired heat loss from the ground
to the atmosphere. Also, it can be seen that decreasing the insulation layer thickness allows for some cooling in winter, but at
the same time, it protects the ground less from heating in summer. The observed effects are plausible because snow accumulates
on top of the thermal insulation in winter, lowering the natural heat loss from the ground. This suggests the existence of an
optimal thermal insulation thickness. However, this optimum naturally depends on the seasonal climatic conditions, as well
as on latitude, altitude, exposition, slope angle and local topography. Given the principal interest of quantitatively assessing
different engineering-based permafrost thermal stabilisation methods, we did not strive to determine this optimum for the



Schilthorn site, as we see this being out of the scope of this study. The brief conclusion here is that the 50 mm thickness provided the better contribution during both summer and winter, compared to the cases with increased thickness (100 mm) and decreased thickness (10 mm, results not shown). However, it can be seen that also this method is not efficient enough for the long-term preservation of local alpine permafrost from the growing impact of atmospheric warming and increasing net energy input into the ground.

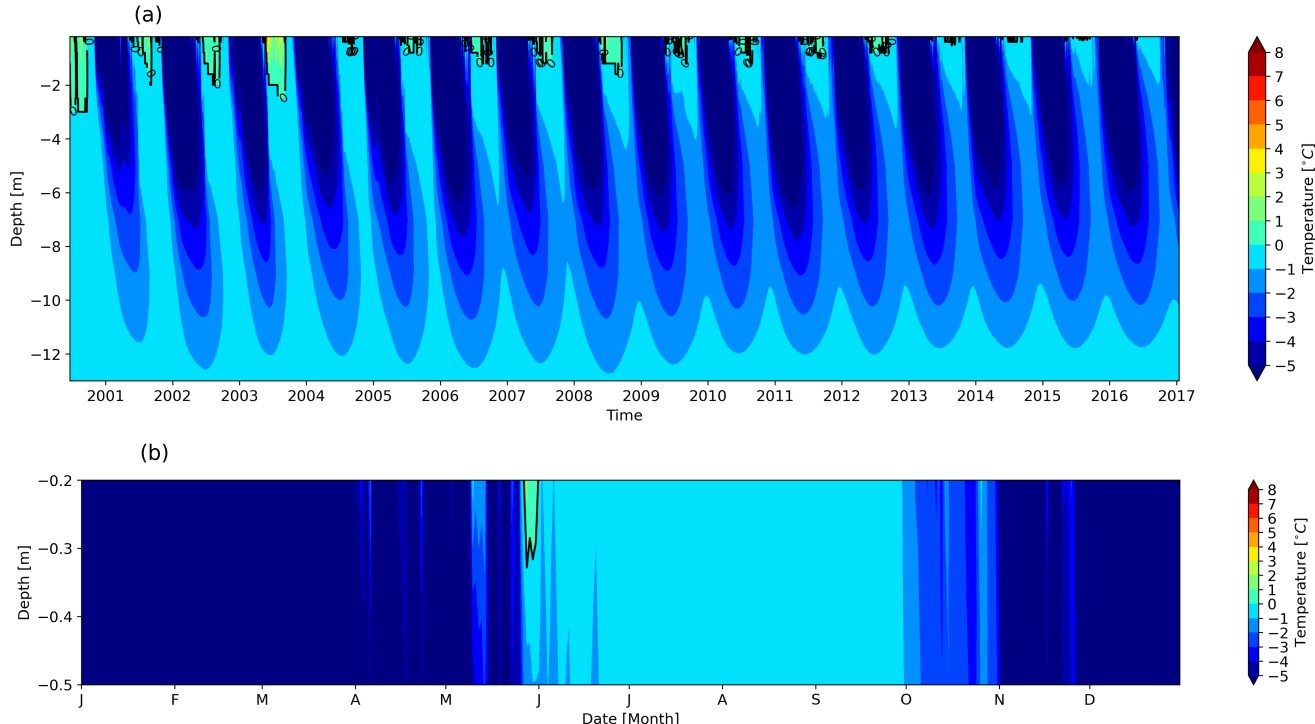

**Figure 7.** (a) Time series of daily averaged modelled ground temperatures from the heat transfer regulation experiment at the Schilthorn site for the period of 2000 to 2017. Black contours indicate the 0 °C isotherm, i.e., the ALT. (b) Last year of the simulation, 2016, after 16 years of thermal ground insulation.

As previously explained, timely placement/lifting of the insulation layer can result in more effective cooling during the winter season and better preservation during the summer season. We examined the impact of heat transfer regulation by combining the impact of thermal insulation with surface shading. Figure 7 shows that with this approach the effect of cooling is intense in winter over a multi-year period favouring heat loss in the ground to a depth of about 10 m (Figure 7a). This method also results in a gradual decrease in ALT over the years, demonstrating that the winter cooling effect surpasses the warming effect of
summer. Additionally, the placement of thermal insulation during summer helps protect the permafrost by reducing penetration of heat from the atmosphere. Taking a closer look at the last year of the simulation (Figure 7b) reveals a temperature decrease in near-surface layers and significant ALT reduction with this approach. Figure B3 indicates the continuous heat extraction




during winter time, explaining the preservation of low temperatures in the ground. This cooling mechanism can be sufficient for creating a thermal (cold) "barrier" layer near the surface to protect the ground beneath over an entire year (Figure 7).

Although the creation of a "barrier" layer takes many years in the simulation, with unstable conditions occurring during its formation, after 16 years the ALT was decreased by several decimetres. We expect this method might be even more effective and used with more benefits in combination with active cooling since it creates an additional "shield" of thermal insulation, which will help to preserve the actively applied cooling at depth while decreasing the heat exchange with the atmosphere.

### 5.2.3 Thermal stabilisation with active cooling

We examine now the effect of the proposed thermal stabilisation system with active cooling (Section 4.2.3)) first quantifying the effect from the cooling pipes alone and second the total effect of the system including the solar panels shading the ground. We examine the temperature and heat flux distribution in the ground to observe if a thermal barrier layer is created.

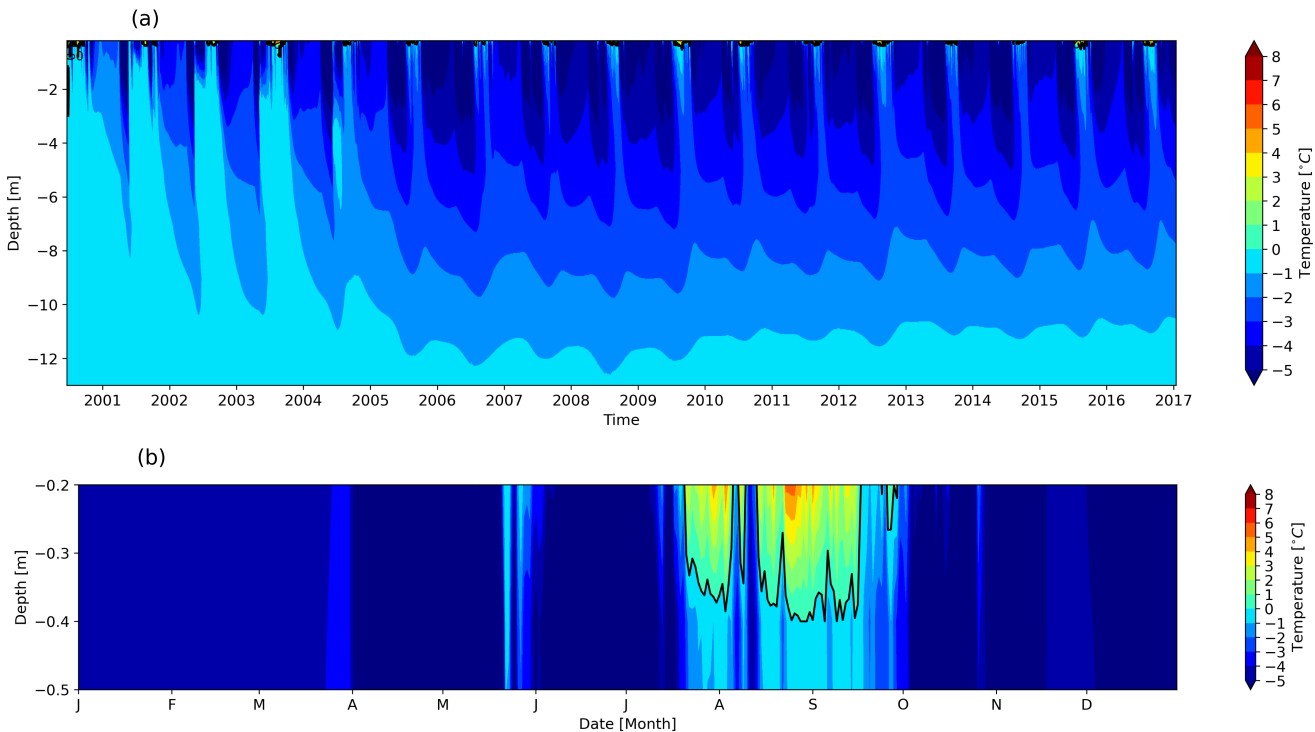

**Figure 8.** (a) Time series of daily averaged modelled ground temperatures from the cooling pipes experiment at the Schilthorn site for the period of 2000 to 2017. Black contours indicate the 0 °C isotherm, i.e., the ALT. (b) Last year of the simulation, 2016, after 16 years of active cooling.





Figure 8 shows that when the active cooling system is in operation, sub-zero temperatures of the permafrost stabilise after
4 years of application of the cooling method with temperatures well below 0 °C at depth. The top layers of the last simulated
year (Figure 8b) show that the thawing boundary, indicated by the 0 °C contour, does not extend deeper than 40 cm. This
demonstrates the effectiveness of the cooling method in creating a thermal barrier layer and limiting heat penetration during
summer.

       The heat flux distribution shown in Figure 9 illustrates the consequences of the formation of a thermal barrier layer. The
large heat fluxes in the upper layers are due to the strong temperature gradient between the air temperature and the cooling
pipes, the latter conserving constant low temperature around them. This determines the direction of the heat flux downwards
from the surface to the level of the pipes. In depth, the heat flux has the opposite sign. Combined, these heat fluxes indicate
heat extraction from the ground via the cooling liquid in the pipes, demonstrating the positive effect of the cooling.

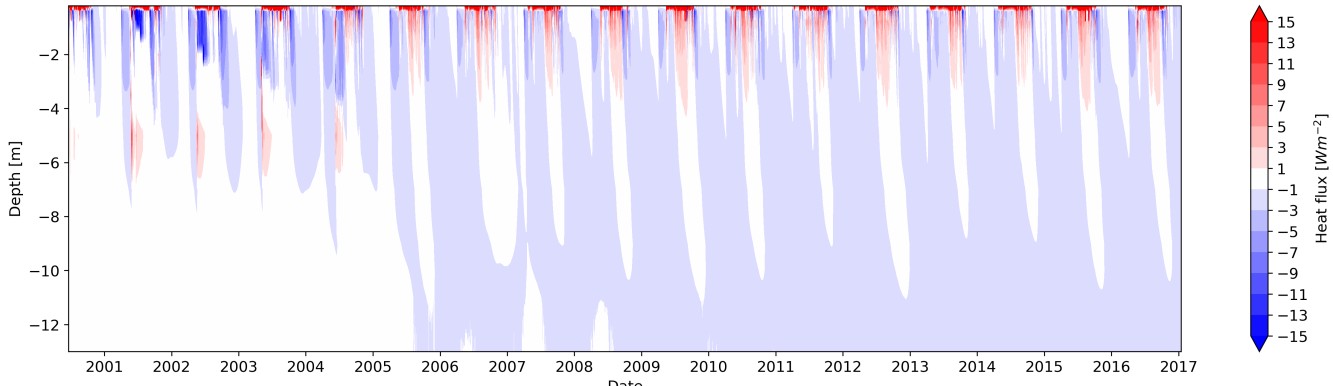

**Figure 9.** Time series of daily averaged modelled conductive heat flux from the cooling pipes experiment at the Schilthorn site for the period
of 2000 to 2017.

Figure 8a shows that after the establishment of the barrier layer (after about 4 years), the soil cools down below a level of
about 3 m, which may be related to the given structural ground stratigraphy (Table 2). The reason why the formation the stable
barrier layer takes several years, is that it take time to establish the energy balance in the ground (Figure 8). This balance is
established when the enough heat is extracted from the ground, and when the purpose of the system is switching from cooling to
keeping the in-depth temperatures stable along the following years. Due to the temperature boundary conditions at the bottom
used in the model (see Section 4.1 the model strives to establish the energy balance between layers, and when equilibrium is
reached the heat exchange between layers is aimed at maintaining this balance (Figure 9). This case with only active cooling,
demonstrates well the reaction of the ground to external active cooling.

       After demonstrating the cooling effect of the pipe system alone, we now focus on the full system combining active and
passive cooling methods, i.e., the ground shading with the solar panels. Simulation results presented in Figure 10 show that
this combined effect is favourable for the conservation of permafrost due to the formation of a more pronounced sub-freezing



thermal barrier layer compared to results from both systems independently. The effect is also more immediate and does not require several years for establishing the thermal barrier layer (Figure 10a). In the final year of the simulation (Figure 10b), it is evident that the impact of summer warming cannot be fully suppressed or compensated. However, the engineered cooling method significantly reduces the ALT compared to natural conditions, achieving a difference of several meters.

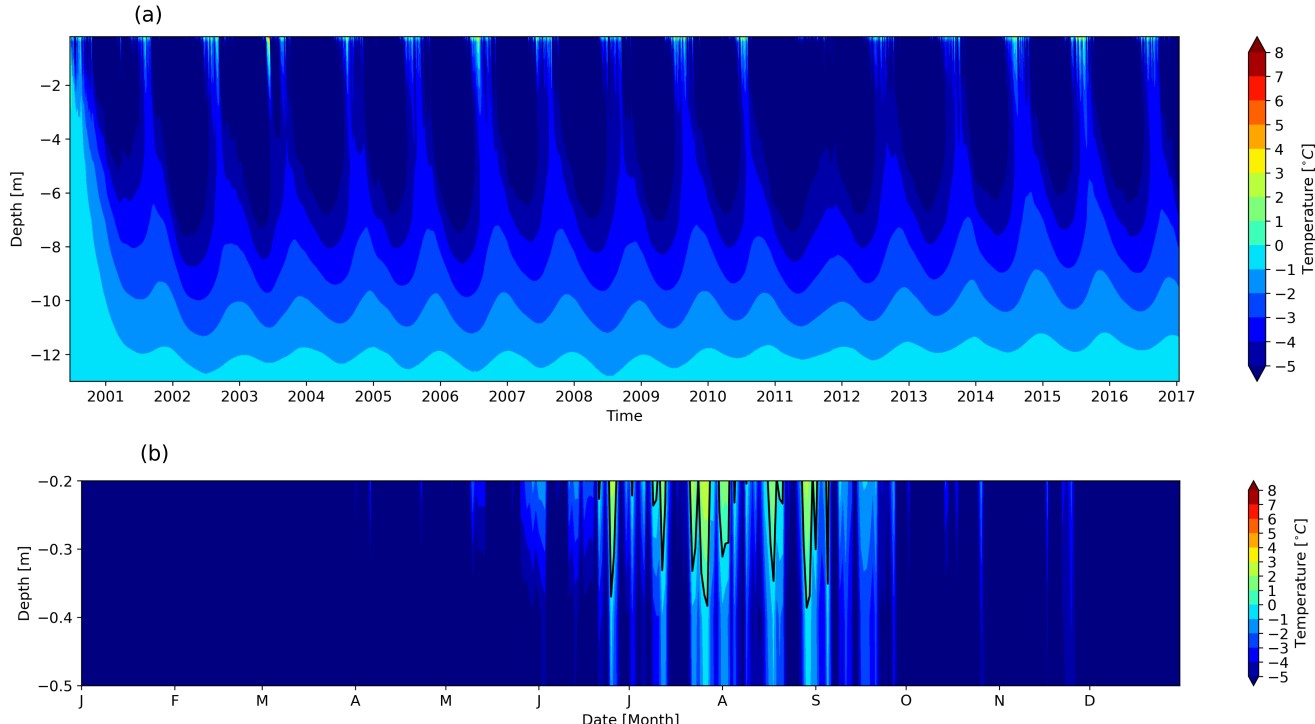

**Figure 10.** (a) Time series of daily averaged modelled ground temperatures from the experiment combining cooling pipes and solar panel shading at the Schilthorn site for the period of 2000 to 2017. Black contours indicate the 0 °C isotherm, i.e., the ALT. (b) Last year of the simulation, 2016, after 16 years of active cooling and shading.

The conductive heat flux evolution instigated by the combined cooling system (Figure 11) shows that in addition to the effect from cooling pipes described before, the passive cooling works in winter season. This corresponds to the heat evacuation from the depth in winter, when the heat flux direction is upwards (Figure 11a). Figure 11b indicates that the highest heat flux values occur from April to the end of October, when the active cooling is applied. This downward heat flux extends to about 10 cm below the cooling pipes, which arises as a result of the strongest temperature gradient. Beneath this level, the heat

flux decreases eventually approaching 0 $\mathrm{Wm^{-2}}$. In winter, heat is lost to the cold atmosphere turning the sign of the heat conduction. Figure C1 presents the daily heat flux and temperature profiles, illustrating the evolution of conductive heat flux and ground temperatures over time and highlighting their dynamic behaviour throughout the simulation.



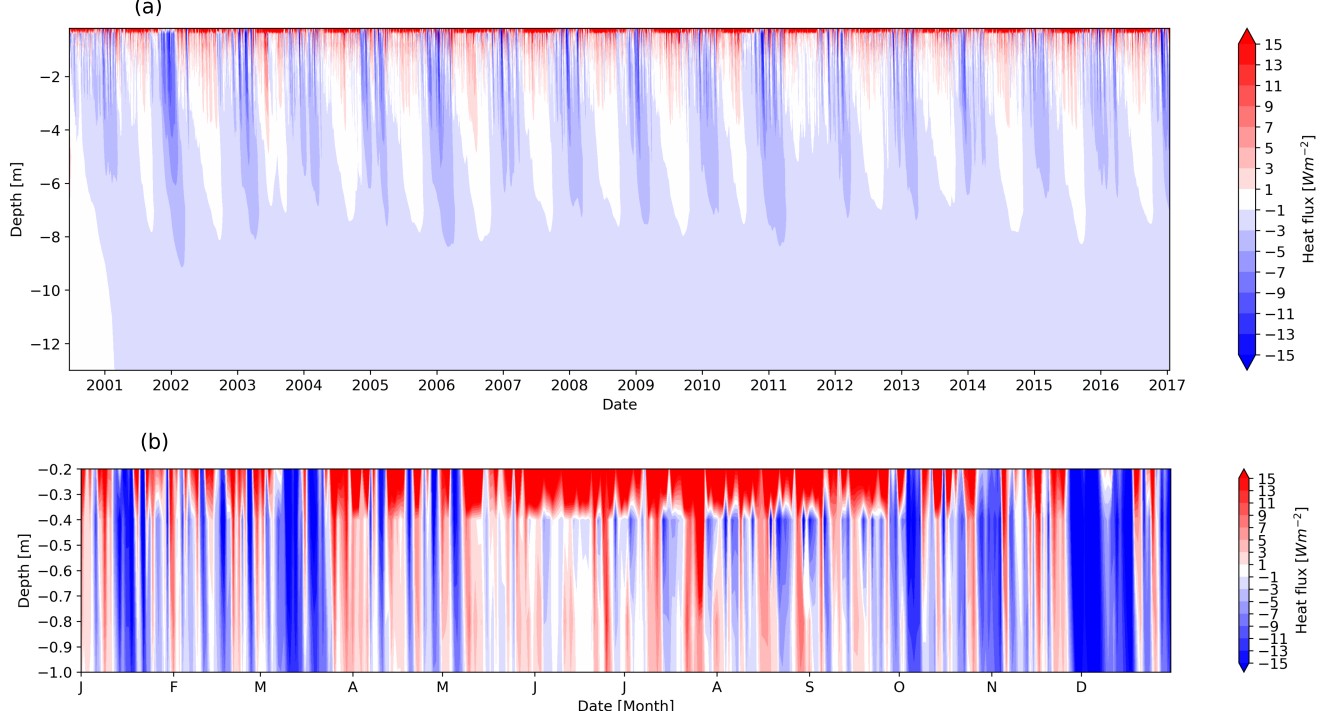

**Figure 11.** Time series of daily averaged modelled conductive heat flux from the experiment combining cooling pipes and solar panel shading at the Schilthorn site for the period of 2000 to 2017. (b) Last year of the simulation, 2016, after 16 years of active cooling and shading.

## 5.3 Analysis of stabilisation effects

For the considered mountain regions, the most efficient method of thermal stabilisation was active cooling combined with
effects from passive methods such as surface shading with solar panels. In the experiments in Section 5.2.3 all solar radiation
was used for powering the heat pump. Despite the fact that the main objective is to set as much energy as needed for the
cooling purpose, we also tested the case, when only a partial amount of available energy from the solar panels is reserved
for the cooling purpose. This case may bring more realistic and practical demonstration of the system application, as it could
imply the potential for using smaller panels or other optimizations in system design. It allows to challenge the system, in
case when the surface for installation is limited and the energy should be redistributed between the cooling and infrastructure
energy supply. We show a situation in which only 50% of the incident solar radiation is used for powering the heat pump,
i.e., for cooling, reserving the other half for the power grid or other needs in the vicinity of the site. As it was explained in
methods (Section 4.2.3), the surface that is cooled is equal to the surface of solar panels. In the following part we made some
experimental tests, with cutting the energy supply directed to cooling by certain amount, to look at the robustness of the system
in different combination.





In the case where only half of the generated PV energy is used for cooling (Figure 12), we applied cooling the entire year, unlike the case discussed in Section 5.2.3, where cooling occurs only from April to October. This energy partitioning between cooling and the grid aims to maintain the same persistence and temperature of the barrier layer as in Figure 10. However, it can be seen that with half the amount of power the barrier layer is not as stable as with 100% energy supply. The ALT is increased when using less power (Figure 12b) compared to the case when using 100% energy for cooling (Figure 10b). Recall, the creation of an efficient barrier layer only with cooling pipes requires 5 full seasons with 100% power supply, applying cooling from April to October. It can be noted that due to the initial spin-up process to create the thermal barrier layer, initially most PV energy is used for cooling the system. Once the barrier layer is temporally stable and established, a larger fraction of the produced PV energy can be injected into the grid. However, it is worth acknowledging that active cooling may not necessarily need to bring the system to a new equilibrium, maintaining the existing thermal balance might be sufficient. It should be noted that our results did not explicitly test this, we assumed a simplified case where the energy is evenly split between cooling and grid injection. In a real application case, the system control will become completely dynamic, i.e., which is considered to be the goal of the system application. It is important to give priority to cooling when required, otherwise feeding into the grid. For the design of a suitable PV system it is therefore imperative to know the energy demand for the creation and maintenance of a thermal barrier layer in alpine permafrost, which can be available from simulations as demonstrated in the previous sections. The optimal choice will enable reliable cooling of the ground while bearing the potential of providing significant excess power to the grid or nearby installation, especially after the initial ramp up time.

Another alternative is the combination of active cooling (Section 4.2.3) with heat flux regulation (Section 4.2.2) as discussed in Section 5.2.2 and presented in Figure 7, when during summer the ground was also protected with thermal insulation material. In that case, there is no need to wait until the snow has melted to lay out the thermal insulation as discussed in Section 5.2.2, however, it is important for the heat flux regulation to follow the active cooling implementation. When the active cooling is applied (from April to October) the thermal insulation is placed on the ground, and when the active cooling is deactivated the thermal insulation should be lifted. This combination helps better conserve the low temperature created by the cooling pipes in the ground during summer, and in winter it allows for the natural cooling process. In this case, a thermal insulation material of 50 mm thickness is used, which was numerically tested here and found to be the most efficient. The use of combination of different methods, the active cooling with shading and heat flux regulation, helps to entirely achieve the stabilization effect. It allows to use less energy from solar panels for cooling and send more to the grid even in case when the cooled surface is equivalent to the surface of solar panels. As in the experiments explained above we tested the different cases when only the certain amount of energy is led to cooling. Here due to effective preservation of cold in the ground established by heat flux regulation, we found that in this specific case for the modelled site, the case of the active cooling in combination with heat flux regulation the energy required for cooling can be reduced to 10% of the one taken from the solar panels.

As the Figure 13 shows the conservation of cold temperature resulting from heat flux regulation with thermal insulation, helps to keep the ALT as shallow as possible. This brings the better cooling and heat extraction from the ground comparing to the cases of active cooling only (Figure 8), or active cooling with shading (Figure 10). Another positive aspect of this combined method is that it ensures not all produced PV energy is spent on cooling only, as it was in the case demonstrated in Figure 10.



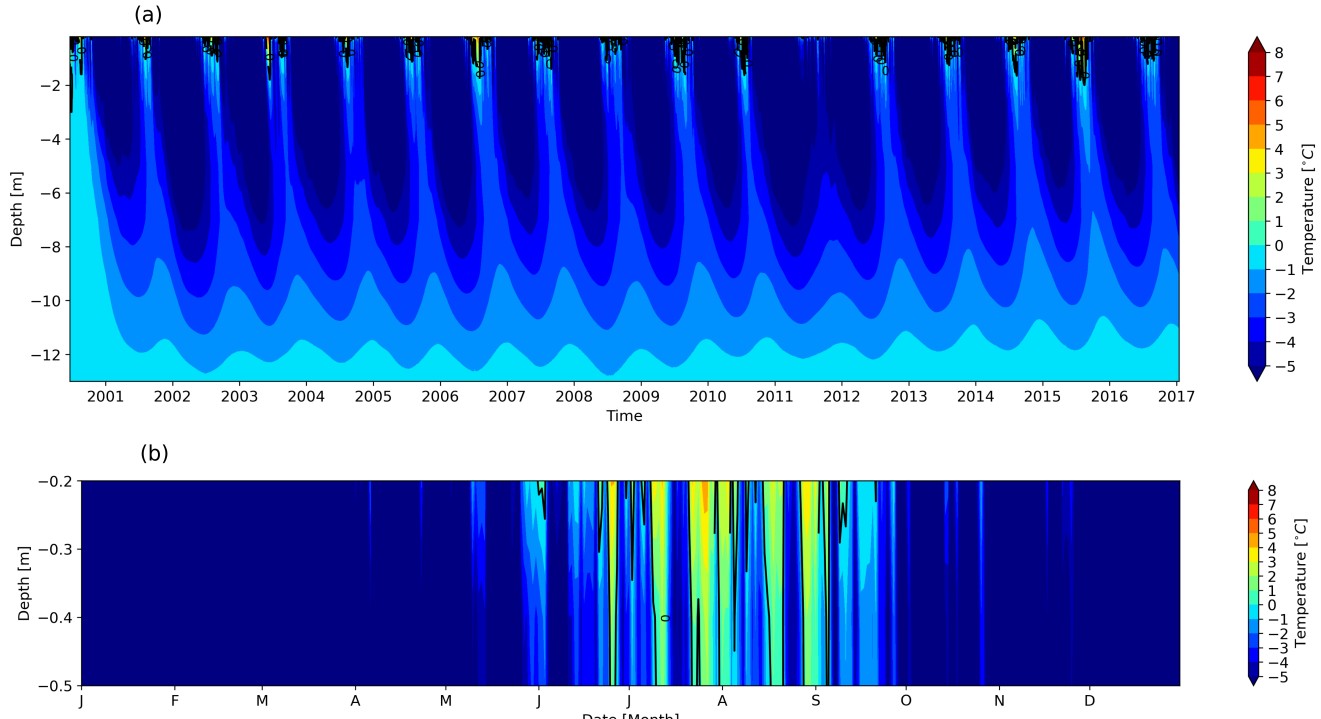

**Figure 12.** (a) Time series of daily averaged modelled ground temperatures from the experiment combining cooling pipes and solar panel shading but using only 50% of available energy for cooling at the Schilthorn site for the period of 2000 to 2017. Black contours indicate the 0 °C isotherm, i.e., the ALT. (b) Last year of the simulation, 2016, after 16 years of active cooling and shading.

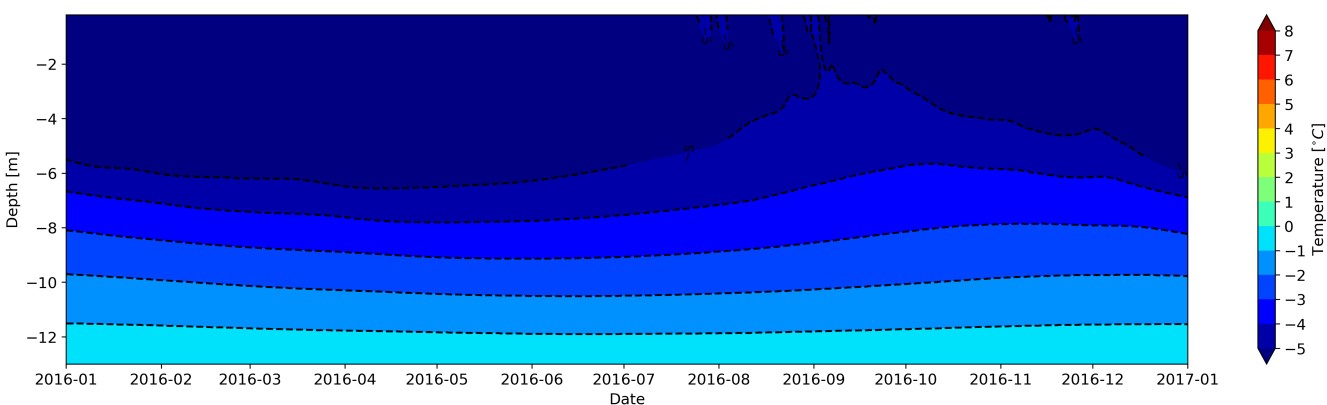

**Figure 13.** Daily averaged modelled temperature from the experiment combining cooling pipes and solar panel shading but using only 10% of available energy for cooling at the Schilthorn showing the last year of the simulation, 2016, after 16 years of active cooling and shading.



As previously discussed, limiting the energy used for cooling without regulating heat flux is possible and recommended to be 50% or higher; otherwise, the active layer will remain deep. When combined with heat flux regulation, this approach offers greater flexibility in managing the power received from solar panels. We also suggest that intense and continuous cooling might be unnecessary, it can be adjusted based on the monitored state of the permafrost, its temperatures, and also based on the atmospheric conditions. The latter determine which combinations of the stabilisation systems should be applied in different climatic conditions.

## 6 Conclusions and outlook

Permafrost thawing causes severe problems such as infrastructure damage or destruction, rock falls, land slides, and other environmental disasters. This study uses numerical simulations to investigate the effect of different engineered mitigation strategies targeted at thermally stabilising permafrost through various passive and active cooling methods. All these methods are geared to artificially cool the ground, limit or favour heat exchange with the atmosphere depending on the season, preserve the permafrost, and minimise the seasonal ALT. One of these methods consists of technically creating and maintaining a cold thermal barrier layer that protects the permafrost underneath from heat conducted from the surface into the ground, particularly during summer when solar radiation input is large. The energy required for the technical cooling of the ground is obtained from installations of collocated solar panels, which are also creating favourable shading of the ground.

A series of numerical experiments assesses several technical solutions or combinations of such thermal stabilisation systems, to demonstrate and quantify their efficiency when applied in alpine permafrost. For the simulations, We use the 1D SNOWPACK model (Section 2) for reproducing the mountain permafrost at a representative alpine location, the Schilthorn site, canton Bern, Switzerland, (Section 3). This site features a time series of observational borehole data for comparison and validation of the model simulations. A first simulation reproduces the natural undisturbed conditions at the Schilthorn site (Section 4.1) configuring the model with parameters of the local substrate measured at that site (Section 5.1). Then, different thermal stabilisation methods were implemented numerically (Section 4.2), to show their ability in cooling the ground and keeping it preserved. Both passive, active, and combined thermal stabilisation systems were tested and their efficiency assessed (Sections 5.2, 5.3). Simulation results indicate that different combinations of thermal stabilisation systems are able to produce the successful effect and to cool the ground sufficiently. However, their efficiency varies depending on the methodology applied.

The tests of passive thermal stabilisation with shading of the surface (Section 4.2.1) do not show the desired long-term effect for thermal stabilisation. It has a good effect during winter, due to natural cooling, however, in summer the ALT stays almost unchanged compared to the natural conditions (Section 5.2.1). Another passive cooling system consists of a thermal insulation layer on top of the ground (Section 4.2.2). In this experiment, two configurations are tested: (a) year-round covering the ground with insulation material and (b) deploying and lifting of the insulation layer depending on the season. The year-round configuration does not result in significant thermal stabilisation, while the season-dependent deployment and removal of a thermal insulation layer effectively regulates the conductive heat flux at the ground surface leading to favourable stabilisation



effects (Section 5.2.2). Seasonal heat flux regulation method of thermal stabilization significantly reduces the ALT, although it
requires several years to fully achieve this effect.

The active cooling system using pipes in the ground (Section 4.2.3) is able to create a stable continuous thermal barrier layer but only after a 4-years spin-up period (Section 5.2.3). The best efficiency of a thermal stabilisation system is reached by combining active and passive methods (Section 5.2.3), which decreases the ALT to only a few tens of centimetres, and results in a stable barrier layer already after the first year. Thus, it is concluded, that the best cooling efficiency is achieved by a tailored
combination of active and passive methods. The combination of active cooling, shading, and the heat flux regulation with the temporary thermal insulation layer which reduces warming and favours cooling has the strongest effect (Section 5.3).

Knowing the amount of energy required for cooling and for creating and maintaining an effective thermal barrier layer to stabilise the underlying permafrost allows for optimal system design and power regulation. Simulation have shown that for an optimal stabilisation system only a fraction of the produced PV energy (50% for only active cooling and 10% when active
cooling is combined with heat flux regulation) is necessary for the direct cooling of the ground. The excess energy is then available for the grid. It is recommended that the applied stabilisation system dynamically regulates the required power supply for ground cooling depending on the monitored state and conditions of the permafrost (Sections 5.2 and 5.3). However, the convincing performance of the combination of active and passive cooling is difficult to achieve in the field owing to complicated installation in complex alpine permafrost terrain. In such cases, the use of a system similar to (Loktionov et al., 2024) could
be an alternative, cooling portions of constructions directly with cooling pipes attached to basement walls taking advantage of unused space in the foundation of a building and avoiding the installation of cooling pipes on inaccessible surfaces such as bedrock.

For a detailed assessment of the thermal stabilisation effects, it is essential to dispose of data of atmospheric conditions such as air temperature, solar radiation, wind speed, phase and quantity of precipitation, and snow depth from nearby meteorological
stations or directly from sensors on-site. This enables a better understanding of the impact of passive thermal stabilisation Section 4.2.1 and Section 5.2.1. Due to the nature of simplified and idealised 1D simulations, i.e., the panels completely protect the ground from the snow and precipitation, not considering the possibility of lateral blowing snow, or leaking liquid precipitation through the gaps of adjacent solar panels Section 5.2, the applied coefficient for wind speed decrease is empirical and its realistic value requires more advanced estimation. In-situ experimental field tests will further advance the study and
understanding of the physical processes resulting from passive cooling due to the shading by the solar panels, and to better quantify the impact of this method.

Further refinement of the system is needed for an optimal choice of system variables such as depth and spacing of the cooling pipes, thickness of a thermal insulation layer, and the most efficient and economic way of creating and maintaining a thermal barrier layer while tuning the power regulation to dynamically adapt to real-time thermal permafrost conditions. The
geometrical configuration of the cooling pipes in the ground needs to be adjusted to the local geo-cryological conditions for sufficient and most efficient ground cooling. Owing to the 1D simulations, such 3D effects could not be investigated in the present study. However, aspects as pipe spacing, diameter, and geometrical layout need to be adjusted to the conditions of the site to ensure optimal cooling as well as maximum solar energy yield at the given site and geographical region. Optimal system



regulations is required for most efficient cooling and for protecting infrastructure at risk. Up-scaling the system to larger cooling areas is another direction of research. At this stage, the numerical modelling could be extended to other permafrost types (continuous - Arctic and Polar zones; discontinuous - Subarctic zone), which will also allow to investigate the performance of thermal stabilisation systems in different regions and to propose an optimal combination of system components for different conditions. The present study constitutes a detailed and systematic numerical investigation for the choice and performance of permafrost stabilisation methods. Results are intended for the design and implementation of prototype installations and for up-scaling to real-world applications. The forthcoming challenges involve conducting experimental tests to validate the hypothesis, evaluating the influence of climate change on permafrost dynamics both with and without stabilisation measures, and devising an optimised methodology for thermal stabilisation tailored to diverse sites and climate conditions.

Despite the efficiency of the cooling system and the resulting successful creation of a barrier layer creating, it is important to note that a few additional aspects which are not considered here have to be taken into account, for instance, the terrain topography, and potential difficulties of building on mountain permafrost. These together with other natural hazard factors may bring up some risks for the infrastructure and cooling system itself. One of the dangers is linked with active cooling and moisture present in the substrate (Section 5.2.3). Intense cooling bears the risk of ice formation in wet ground, which may lead to heave and potential destruction of solar panels and cooling pipes, and damages to nearby constructions which are supposed to be protected. That is why for specific applications the simulated case must be adapted according to the local ground properties and moisture conditions, the existing ground temperature distribution, ground-atmosphere heat exchange, topography, exposition, local climate, and potentially existing construction characteristics. It remains a challenge to apply a thermal stabilisation system in alpine permafrost without altering the structure and mechanical processes in the ground.

This study shows an example of the implications faced by infrastructure built on the mountain permafrost. With permafrost temperatures rising, the stability of infrastructure is increasingly at risk (Duvillard et al., 2021). Currently the methods of monitoring and mitigation in mountain permafrost regions are facing some challenges in an adaptation to the warming processes and require enhancement to adapt to the rapidly changing conditions, and delay further thawing, and increasing of ALT (Haeberli et al., 2023). Our study focuses on a mountain permafrost site Schilthorn in Switzerland, identified as one of the most vulnerable locations, where the ALT has doubled over the past decade (Hauck and Hilbich, 2024). We showed the ability of thermal stabilization methods to delay thawing, protecting infrastructure with adaptable and scalable cooling techniques. This study on mountain permafrost demonstrates that thermal stabilization methods perform just as effectively in mountain permafrost as they do in lowland permafrost zones (Loktionov et al., 2022). Given their comparable success in different soil types, these methods can be broadly applied to alpine permafrost, extending beyond the specific site studied.

*Code and data availability.* Data and code used in this study will be published after acceptance of the paper but are available to reviewers upon request.



## Appendix A: Natural conditions

### A1  Snow height

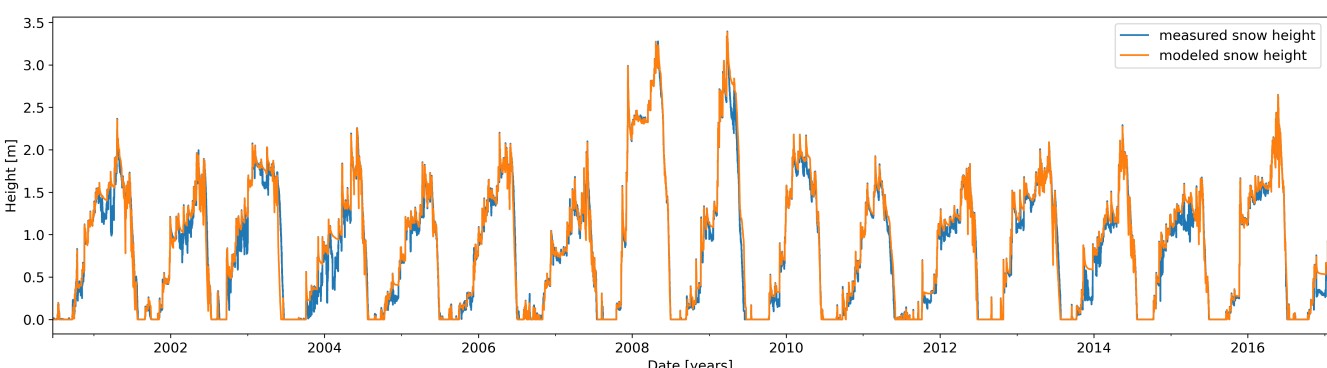

**Figure A1.** Time series of daily averaged measured and modelled snow height during natural undisturbed conditions at the Schilthorn site from 2000 to 2017.

### A2  Ground temperatures

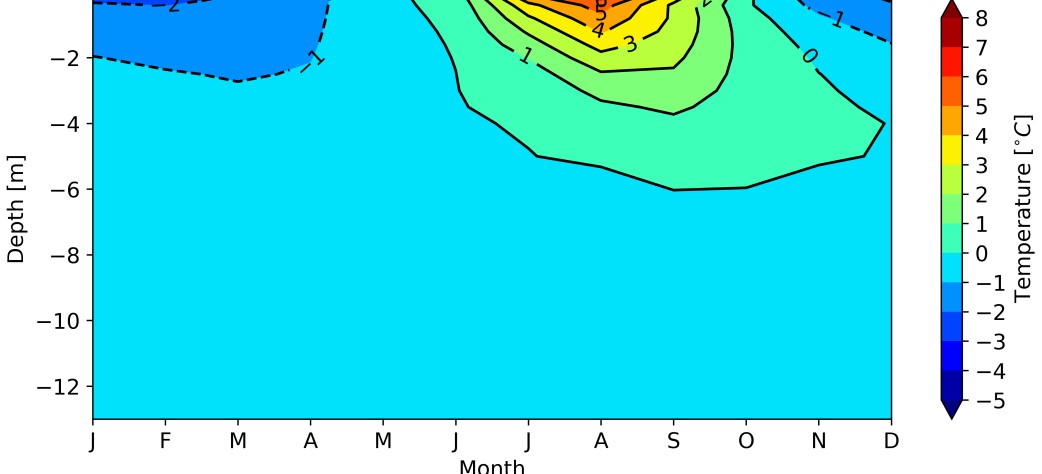

**Figure A2.** Monthly averaged modelled ground temperatures at the Schilthorn site from 2000 to 2017 resulting from natural undisturbed conditions at the Schilthorn site from 2000 to 2017.




## Appendix B: Thermal stabilization with thermal insulation

### B1    Thermal insulation conductive heat flux distribution

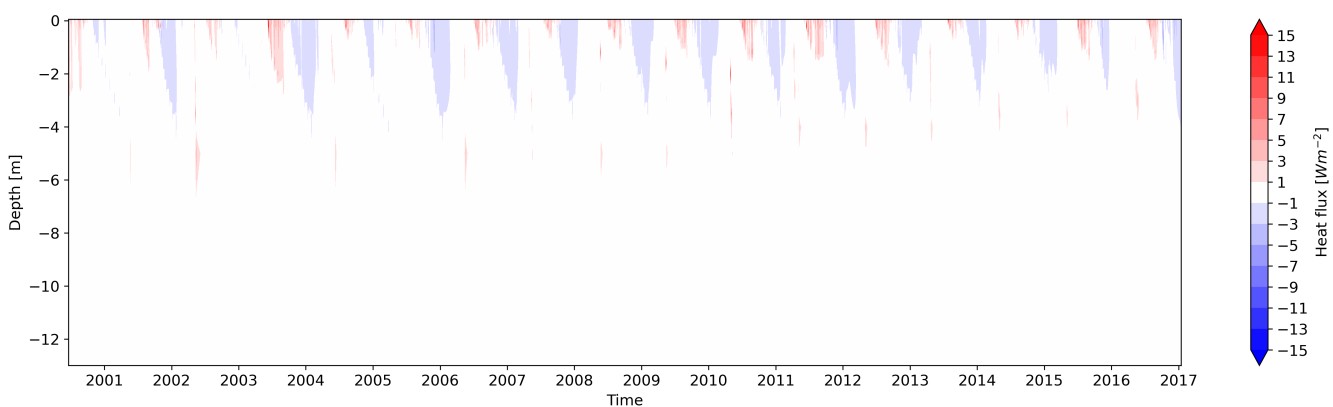

**Figure B1.** Time series of daily averaged modelled conductive heat flux from thermal insulation experiment using 50 mm thermal insulation layer at the Schilthorn site for the period of 2000 to 2017.

### B2    Thermal insulation thickness effect

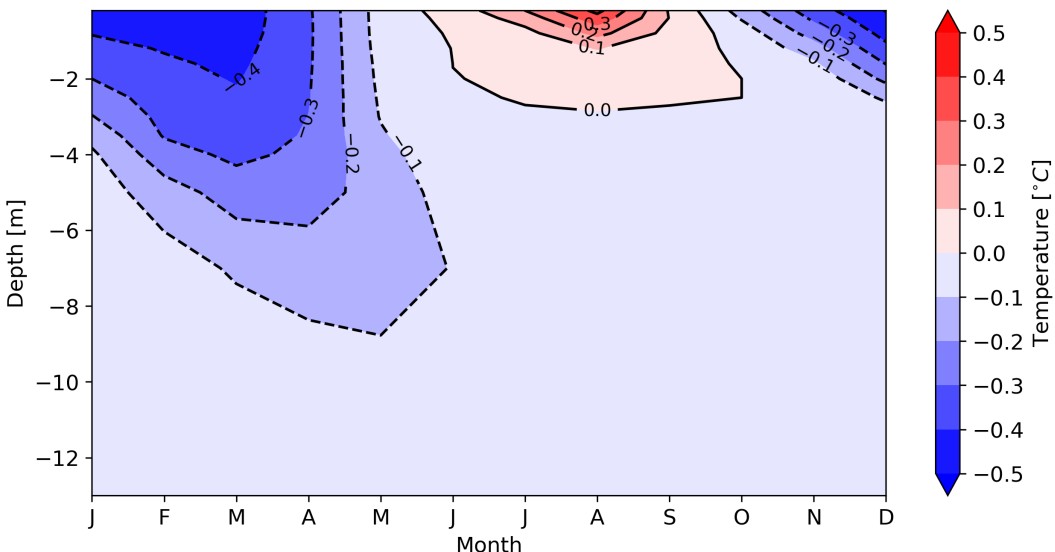

**Figure B2.** Monthly averaged temperature differences between 50 mm and 100 mm (50 mm - 100 mm) thermal insulation thicknesses.



## B3    Heat flux regulation distribution of conductive heat fluxes

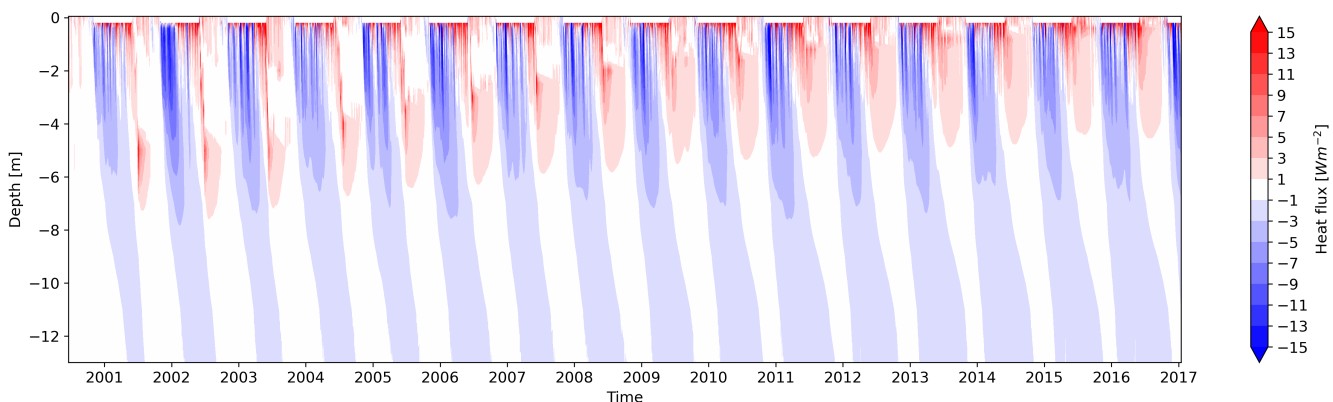

**Figure B3.** Time series of daily averaged modelled conductive heat flux from the heat transfer regulation experiment at the Schilthorn site for the period of 2000 to 2017.

## Appendix C:  Thermal stabilization with active cooling

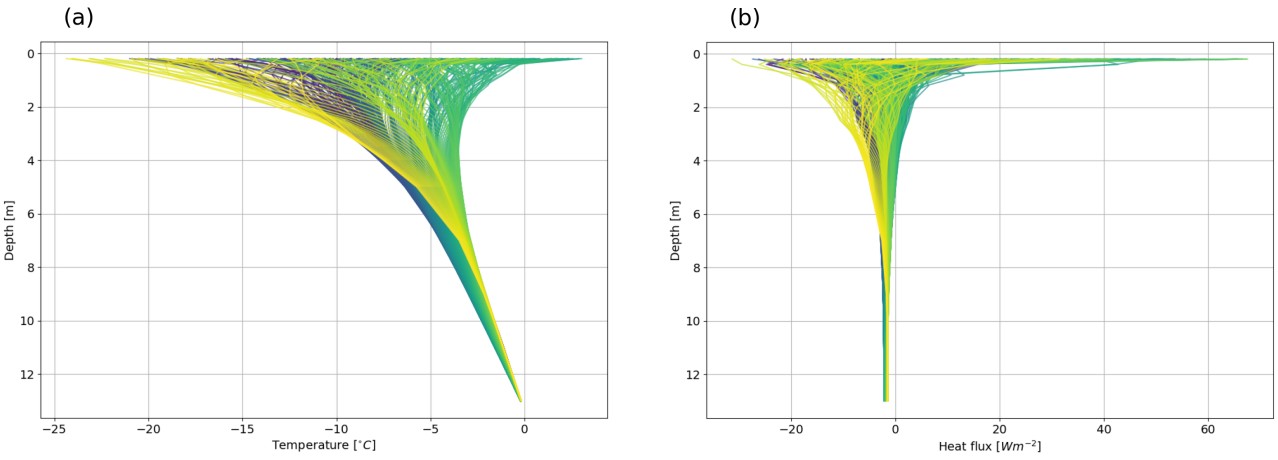

**Figure C1.** (a) Daily averaged temperature distribution, and (b) daily averaged heat flux distribution for the last year of the simulations, 2016 after 16 years of active cooling and shading.



*Author contributions.* ES, ML and HH designed the research. ES prepared the data, run the simulations, analysed results and wrote the draft. ML, NW, MP, HH contributed to the results and data analysis and revised the manuscript. ML and HH supervised the research. NW
maintained and corrected the SNOWPACK code. MP provided expertise in permafrost processes. All authors contributed to the final version of the manuscript.

*Competing interests.* The corresponding author has declared that none of the authors have any known competing financial interests or personal relationships that could have influenced the work reported in this paper.

*Acknowledgements.* We acknowledge MeteoSwiss and IMIS for providing the observational data used in this research. We acknowledge the
600 editor Prof. Hanna Lee for valuable feedback and insightful comments, which contributed to the improvement of this manuscript.



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
