# Peer review of "Assessment of thermal stabilisation measures based on numerical simulations at a Swiss Alpine permafrost site"

_EGUsphere, 2024_

## Author Response (AR1)

The authors thank the reviewers for the comments and feedback. Here we list the specific changes made in manuscript:

the replies to the first review can be found by link:
https://doi.org/10.5194/egusphere-2024-4174-AC1

 the replies to the second review can be found by link:
https://doi.org/10.5194/egusphere-2024-4174-AC2

Additionally, we have made some changes in the thickness of the gridcells, mentioned in the second review (*Table 2: Are the 3 layers divided into gridcells? What is the thickness of the gridcells and do they vary with depth?*): When simulating the thermal stabilization with thermal insulation the thermal insulation layer and layers between 0 m and 0.2 m were divided into grid cells of 0.05 m for both 50 mm and 100 mm material thickness. For thermal stabilization with active cooling to accurately represent the presence of cooling pipes, the grid cell thickness for layers between 0 m and 0.8 m was refined to 0.005 m.

To make it easier to navigate the corrections inlined with the review, we have prepared a table listing the line numbers with comments and their corresponding corrected lines in the tracked changes file:

| Review 1 | | Review 2 | |
|---|---|---|---|
| Old line number and comment | New line number with correction in the Track changes file | Old line number and comment | New line number with correction in the Track changes file |
| L190: The observational permafrost temperature and atmospheric data sets of this site are largely sufficient -> "largely" based on what? | L264 | L23: Include newer rates of Noetzli et al. (2024): Enhanced warming of European mountain permafrost in the early 21st century, https://doi.org/10.1038/s41467-024-54831-9 | L25-L27 |
| L219: The simulations have been run from June 2000 to January 2017-> Why this period? Explain, please. | L307-L308 | L36: Not only thawing but already warming permafrost can be a risk for the infrastructure. L36: Why "built" infrastructure? | L44 |
| L220: …an hourly time step-> Why hourly? Explain, please. | L315-L320 | L37: What is meant with "such" infrastructure? Infrastructure on permafrost? | L44-L47 |
| L265: The objective of this experiment is to allow more efficient… -> Is that really the objective? It is not to analyze the applicability? | We removed this sentence. The objectives have been regrouped and reformulated at the end of the Introduction section.(L162-L173) | L49: What includes "other destructions"? | L59-L61 |
| L268: We simulate the presence of a 50-100 mm thick… -> Why that thickness and not a different one? Explain, please. | L390-L396 | L54: Improve the clarity of this sentence. | L70-L77 |
| L273: The albedo of the isolation material is assumed to be 70%… -> 70% based on what? Explain, please. | L399-L401 | L108: also CryoGrid community model, https://doi.org/10.5194/gmd-16-2607-2023 | L140-L144 |

| | | | |
|---|---|---|---|
| L299: …as evident in Equation (1).-> Not sure that "evident" is the right choice. If something is evident, there is no need to say it. | L432 | L138: Did you include drainage / seepage in combination to your bucket water scheme? Your simulated fieldsite is on a slope (Fig. 1), so this might be an important effect. If not included, discuss it in the uncertainties. | L790-L794 |
| L329: For some periods, measured snow depth is below the simulated one indicating that the model would underestimate melt or erosion.-> What is the reason for this? Can you support this statement with a reference? | L470-L471 | L166: Not clear, so it calculates the surface energy balance? What does it mean that it avoids the connection with the surface temperature? | L222-L227 |
| | | L174: Which permafrost parameters? Ground temperature and active layer thickness? | L232-L235 |
| | | L175: How is the snow simulated? Based on the precipitation of the atmospheric data? On a mountain top that might be extensive snow redistribution due to wind. Is that considered? Furthermore, you are on sloping terrain, which may affect the snow accumulation. Do you take this into account? How is the melting handled? Is meltwater just removed from the system or can it infiltrate the ground? | L236-L239 |
| | | L178: This sentence does not make sense. Do you mean: We selected this site because it us representative and because of the infrastructure? | L247-L249 |
| | | L180: Give time period and active layer thickness before and after. What about the variability of ALT from year to year? | L251-L253 |
| | | L193: Atmospheric data from which time period? Also refer here to table 1, as it is only listed in this table which parameters have been used and not mentioned in the text. | L270-L274 |
| | | L198: How was the spinup performed? | L290-L292 |
| | | L199: Is this data from the borehole? Then add it to table 1 where you describe the borehole. and L205: You say | L281-L284 |

| | | | |
|---|---|---|---|
| | | in line 199 that you have observational data for volumetric ice and water content, voids, density, thermal conductivity and heat capacity. Now you say it is based on borehole temperatures (?) and modelling. Which is true? | |
| | | Table 2: Are the 3 layers divided into gridcells? What is the thickness of the gridcells and do they vary with depth? | Each layer was divided into grid cells with a vertical extent of 0.1 m.(L291-L292)

Similarly to the ground layers (Section 3), the thermal insulation layer and layers between 0 m and 0.2 m were divided into grid cells of 0.05 m for both 50 mm and 100 mm material thickness. (L397-L399)

Advective cooling was applied for the months of April to October. The cooling pipes are integrated into the model in a layer from 20 cm to 22.5 cm (the diameter of the pipe is 25 mm), cooling this layer to a minimum temperature of −7.5 ◦ C, which corresponds to the mean temperature of the coolant circulating in the pipes (Loktionov et al., 2024a). To accurately represent this process, the grid cell thickness for layers between 0 m and 0.8 m was refined to 0.005 m. (L441-L445) |
| | | L226: For those not familiar with Schmucki, please state the main principle so that one does not have to google the paper: is it e.g. an albedo aging factor? | L326-L329 |
| | | L253: Partly based? Which of the numbers were not based on Loktionov et al., 2022? | L372-L378 |
| | | L309: Where does the number | L443-L444 |

| | | | |
|---|---|---|---|
| | | -7.5 °C come from? | |
| | | L329: What about redistribution of snow by wind? | L332-L333 |
| | | L342: Could it be that this is true because you used the bucket water scheme? Do you think your results could look different using Richards equation? Discuss the uncertainties. | L786-L794 |
| | | Fig. 4b / L343: In your simulations, reduced wind speeds lead during the entire year to lower ground temperatures. I am wondering: if you reduce wind speeds, this will reduce latent heat fluxes during summer, decreasing the evaporation and thus the cooling of the ground? Or is evaporation not included in the model? | L795-L798 |
| | | L350: "affects negatively ground temperatures" can be misleading as temperatures are increased. Change the wording. | L492-L495 |
| | | | |